# Spinal Cord Injury and Complications Related to Neuraxial Anaesthesia Procedures: A Systematic Review

**DOI:** 10.3390/ijms24054665

**Published:** 2023-02-28

**Authors:** Daniel H. Pozza, Isaura Tavares, Célia Duarte Cruz, Sara Fonseca

**Affiliations:** 1Experimental Biology Unit, Department of Biomedicine, Faculty of Medicine of Porto, University of Porto, 4200-319 Porto, Portugal; 2Institute for Research and Innovation in Health and IBMC (i3S), University of Porto, 4200-135 Porto, Portugal; 3Anaesthesiology Department, São João University Hospital Centre, 4200-135 Porto, Portugal

**Keywords:** spinal cord injury, spinal anaesthesia, epidural anaesthesia, anaesthesia, analgesia, paraplegia, hematoma, acute back pain, sensorial deficit, motor deficit, neuraxial technique

## Abstract

The use of neuraxial procedures, such as spinal and epidural anaesthesia, has been linked to some possible complications. In addition, spinal cord injuries due to anaesthetic practice (Anaes-SCI) are rare events but remain a significant concern for many patients undergoing surgery. This systematic review aimed to identify high-risk patients summarise the causes, consequences, and management/recommendations of SCI due to neuraxial techniques in anaesthesia. A comprehensive search of the literature was conducted in accordance with Cochrane recommendations, and inclusion criteria were applied to identify relevant studies. From the 384 studies initially screened, 31 were critically appraised, and the data were extracted and analysed. The results of this review suggest that the main risk factors reported were extremes of age, obesity, and diabetes. Anaes-SCI was reported as a consequence of hematoma, trauma, abscess, ischemia, and infarction, among others. As a result, mainly motor deficits, sensory loss, and pain were reported. Many authors reported delayed treatments to resolve Anaes-SCI. Despite the potential complications, neuraxial techniques are still one of the best options for opioid-sparing pain prevention and management, reducing patients’ morbidity, improving outcomes, reducing the length of hospital stay, and pain chronification, with a consequent economic benefit. The main findings of this review highlight the importance of careful patient management and close monitoring during neuraxial anaesthesia procedures to minimise the risk of spinal cord injury and complications.

## 1. Introduction

Spinal cord injuries (SCI) due to anaesthetic practises are rare events but remain an important concern for many patients undergoing surgery. The prognosis of anaesthesia-associated SCI (Anaes-SCI) is devastating, with a presumed mortality risk associated. Moreover, this type of SCI may cause long-lasting effects with severe consequences for the quality of life of affected individuals. Morbidities associated with Anaes-SCI include transient or permanent neurological symptoms, epidural haematoma, or abscess (often associated with irreversible neurological changes, such as paresis, if not diagnosed and treated in a timely manner), direct traumatic spinal injury and adhesive arachnoiditis. They all may be accompanied with pain (back pain), paraesthesia, hypoesthesia, or even permanent anaesthesia and/or motor deficits [1,2,3,4,5,6,7,8,9,10,11,12,13].

Anaesthetic procedures often involve neuraxial techniques, including epidural and spinal [14]. Epidural is a frequently used technique that effectively provides pain relief during and after surgery. It is also used for pain management after trauma and in critically ill patients [2,15,16]. When compared to an epidural, the spinal technique was reported to be easier, faster, and more reliable [5]. Additionally, it was associated with significantly fewer complications when compared to epidural or combined approaches [17]. This is probably related to the technique per se, since the subarachnoid technique is generally used as a single shot procedure for anaesthesia and the epidural technique is used as a continuous technique mainly for intra- and/or post-operative analgesia, therefore a catheter remains in place.

In order to prevent complications, a careful preoperative interview and physical examination of patients are usually performed to identify clinical situations that increase the risk of complications associated with the neuroaxis technique approach. This is particularly difficult for traumatic SCI patients receiving care in the emergency room, as they may already present some degree of tissue damage, including laceration of the meninges and neuronal tissue [6,18,19,20].

Although epidural technique is considered relatively safe, patients with spinal canal malformations, extremes of age, immunocompromised or critically ill, are at high risk of Anaes-SCI [1,2,3,5,7,16,21,22,23,24]. Likewise, polytraumatized patients with previous neurological disease, pregnant patients with spinal pathology, patients submitted to antiaggregating or hypercoagulation therapies, or patients presenting abnormal vascular supply or neurological deficits are also at higher risk [1,2,3,5,7,16,21,22,23,24]. In patients with traumatic spinal cord injury, neuraxial technique is generally not recommended. Due to the risks of fluctuation of blood pressure or other signs of autonomic hyperreflexia, special care should be taken [5,25,26,27,28].

Despite the low reported frequency of Anaes-SCI but considering its high-risk consequences to patients, it is important to critically gather and analyse data concerning this type of SCI. Therefore, the present systematic review aims to characterize high risk patients, summarize causes, consequences, and management/recommendations of SCI due to neuraxial techniques.

## 2. Materials and Methods

The present research was conducted in accordance with the Cochrane recommendations on systematic reviews and followed the Preferred Reporting Items for Systematic Reviews and Meta-Analyses (PRISMA) guidelines [29,30]. The review protocol was preregistered in the International Prospective Register of Systematic Reviews (PROSPERO) with No. 378214. The electronic databases used were PubMed, Scopus, and Web of Science. The article search was performed by two independent researchers starting in 12 October and finishing in 18 November 2022. Additionally, the other two authors of this manuscript reviewed all the included manuscripts, and a consensus was reached.

For the present review, we used the patient, intervention, comparison, and outcome (PICO) strategy, and the question was: “What are the causes, consequences, and management/recommendations of spinal cord injury due to neuraxial techniques anaesthesia human patients?” The following Mesh terms were used in the PubMed research: “Spinal Cord Injuries” [Mesh] AND “Anaesthesia and Analgesia” [Mesh]. For Web of Science, the Keywords were: “Spinal Cord Injuries due to Anaesthesia.” Finally, the Scopus search for articles used: “Spinal Cord Injuries” and “Anaesthesia.” Additional search was performed in PubMed using: “Spinal Cord Injuries” [Mesh] AND “Anaesthesia and Analgesia” [Mesh] AND Paralysis; “Spinal Cord Injuries” [Mesh] AND “Anaesthesia and Analgesia” [Mesh] AND toxicity; “Spinal Cord Injuries” [Mesh] AND “Anaesthesia and Analgesia” [Mesh] AND dysesthesia; “Spinal Cord Injuries” [Mesh] AND “Anaesthesia and Analgesia” [Mesh] AND hematoma; “Spinal Cord Injuries” [Mesh] AND “Anaesthesia and Analgesia” [Mesh] AND Awakening; “Spinal Cord Injuries” [Mesh] AND “Anaesthesia” [Mesh] AND neuropathy. A manual search of articles was also performed by the authors to ensure the maximum finding for manuscripts.

Screening of articles was conducted by all the authors to determine eligible studies based on the inclusion criteria: publications in the last 40 years, including only case reports or case series, and epidemiological/clinical studies written in English. Exclusion criteria: comments and editorials, only general anaesthesia, complications related to the spinal cord from a previous injury not related to the anaesthetic procedure, lesions caused by other needling causes such as acupuncture or the treatment of chronic pain, and metastasis in the spinal cord due to cancer discovered during anaesthetic techniques. The selection process followed the PRISMA guidelines [31] and is depicted in Figure 1.

Data extraction from the articles comprised the type of study, type of anaesthesia, causes of spinal cord injury, complications, treatments, and recommendations. The risk of bias was not assessed since all the studies were case reports and comprised only cause-consequences of spinal cord injuries. The manuscript selection was performed with the agreement of all authors.

## 3. Results

The search for publications resulted in 384 manuscripts: 131 in PubMed, 184 in Web of Science, 59 in Scopus, and 10 on additional research. After reading titles and abstracts, 54 manuscripts were initially selected. Duplicate manuscripts were excluded, resulting in 50 articles being included for full text evaluation. Subsequently, 19 studies were further excluded due to several motives: two articles were removed because they were related to patients who already had SCI due to other causes than anaesthesia-related causes: three were related to complications of general anaesthesia; six were published in languages other than English; one reported interscalene block during general anaesthesia; one was a review of the literature and did not include any case reports; five articles were removed since the cause of SCI was tumour-related; and one had been published more than 40 years ago. Thus, 31 manuscripts were included in this systematic review for comparison.

The first analysis evaluated 20 single case reports and seven series of cases (totalling 20 patients), comprising a total of 40 patients with Anaes-SCI. The most commonly used anaesthetic technique was epidural (29 patients), followed by spinal (9 patients), and combined epidural and spinal anaesthesia (1 patient). In one case report, the anaesthetic procedure was not reported. The neuraxial procedures were associated with general anaesthesia in 19 cases. Twenty-three patients were punctured in the lumbar region, 12 in the thoracic, four in the cervical, and three in the thoracic-lumbar, and in two cases, the placement level was unknown/not reported.

The main risk factors reported were extremes of age (1 child, 6 late elderlies, and 6 early elderlies) and the presence of obesity and/or diabetes (2 obese and 2 diabetic). The possible reasons/aetiology of Anaes-SCI were: hematoma (14 cases), unspecified catheter/needle trauma (7 cases), abscess (5 cases), ischemia (4 cases), infarction (3 cases), adhesive arachnoiditis (2 cases), haematomyelia (1 case), unspecified inflammation (1 case), and not reported/unknown (5 cases).

As a result, motor deficits were reported in several patients. Paraplegia was reported in 27 patients, while dyskinesia, or motor weakness, was observed in seven patients. The most commonly reported symptoms were sensory loss (20 patients) and pain (9 patients). Urethral sphincter tone absence and/or urinary incontinence were reported by five patients. There were also four deaths reported (hypotensive crisis [9], massive pulmonary embolus [23], septic shock [32], and ischaemic cerebellar stroke [33]), during or after the management of the hematomas/injuries).

Many authors reported delayed treatments to resolve Anaes-SCI, that included 17 surgeries for hematoma decompression/laminectomy, catheter removal in 5 cases, and rehabilitation for 10 patients. The most commonly used drugs were corticoids to reduce inflammation and antibiotics in abscess cases. The main findings are summarised in Table 1.

The second analysis included two prospective and two retrospective studies, comprising 41,251 patients who received neuraxial block. One manuscript also evaluated the peripheral nerve block [13]. The most frequent complication was localised pain in 9.05% of the cases, followed by 3.1% of sanguineous punctures. Adverse neurological outcomes affected 1.12% of the patients, and 0.08% had anaesthetic toxicity or permanent peripheral nerve injury after neuraxial block. Epidural haematoma frequency was between 0.03% and 0.02%. Finally, 0.03% of epidural abscesses were reported. One study focused in 9 cases of epidural abscesses with important negative consequence such as lower-limb paraplegia, urinary or faecal incontinence, or irradiating pain.

The analysed manuscripts recommended improved anaesthetic procedures, the need to be aware of risk patients, and stressed the importance of early diagnosis combined with proper treatment and, whenever possible, the support of acute pain units for the Anaes-SCI management (Table 2).

## 4. Discussion

Our search clearly demonstrated that, despite being a very rare unfortunate complication, Anaes-SCI are associated with detrimental and untreatable consequences, including paraplegia and death. Most of the minor Anaes-SCI will be resolved in the first 6 months, but it should still be taken cautiously considering the devastating consequences for patients, families/caregivers, and anaesthesiologists [3,5,6,9,10,52]. Due to safety concerns, in the absence of absolute contraindications, neuraxial anaesthesia is often preferred over general anaesthesia in critically ill patients. While there are some deaths that may be directly or indirectly related to the anaesthesia procedure, it is commonly acknowledged that the underlying critical or advanced illness is the primary cause of death in most cases [9,23,32,33,52].

The frequency of spinal-epidural hematoma, ischemia, abscess, or meningitis leading to SCIs due to neuroaxis anaesthesia is reported in less than 0.03% of patients [13,16,17,52,53]. While major complications can occur in up to 1.5% of the patients, other minor complications, such as localised pain at the epidural insertion, are reported by 9% of the patients and are related mainly to multiple block attempts and poor post-operative patient-controlled epidural analgesia [50]. It has also been reported that the risk of a sanguineous puncture increases with patient age and is related to the puncture level, with a higher risk in more caudal segments [51]. Advanced age also increases the risk for dural perforation, while the size of the patient is related to the risk of catheter misplacement, being higher in shorter individuals [51]. A retrospective study reported that 11% of patients submitted to neuraxial block presented side effects or complications, including sensory or motor deficits, nausea or vomiting, and pruritus [13]. Permanent peripheral nerve injury, subcutaneous cell tissue hematoma, epidural abscesses, and arachnoiditis have also been reported, affecting less than 0.1% of the cases [13].

Due to its infrequency, underreporting, and bias in insurance-based data, the causality of post-operative neurological deficits or exacerbation of pre-existing neurological disorders makes it extremely difficult to obtain reliable and consistent information about Anaes-SCI. It is believed that clinical studies potentially underestimate the true incidence, and regional anaesthesia is easily blamed [5,6]. In our systematic review, only a few case reports were found in the literature, and it is clear that anaesthesiologists should be motivated to increase reporting of Anaes-SCI to prompt technical improvement to avoid this type of SCI and treat its consequences.

There is a claim to include the risk of permanent neuropathy from neuraxial block techniques in the informed consent discussions with patients, mainly for high-risk patients such as those with pre-existing neurologic disorders, immunocompromised status, diabetes mellitus, high weight and body mass index, a lower spine approach, antiaggregating or hypocoagulated patients, extremes of age, and critical care patients [6,13,16,23,34,43,51,54,55,56]. In addition to human error, other risk factors may occur since serious injuries also occur in healthy patients receiving competent care. These risk factors are not always known to the anaesthesiologist, making a high proportion of Anaes-SCIs not entirely predictable or preventable [6,19]. Frequently, there is no clinical or radiographic evidence of direct trauma, leaving no clear explanation of Anest-SCI aetiology. In these conditions, the diagnosis of Anaes-SCI is only made after the development of neurologic disturbances [8,10,36], including paraplegia that can result from spinal cord compression, infarction, or direct trauma. In fact, the causes of Anaes-SCI were various and sometimes combined mechanical, ischemic, and neurotoxic insults, vertebral canal abscess, hematoma, meningitis, nerve injury, and adhesive arachnoiditis [10,13,16,32,33,38,39,40,46,47,48,49,50,57]. These insults can lead to numbness and weakness [44,46], total spinal anaesthesia due to nerve blocks [58], pain [16,43,50], paraparesis [16,38], reversible paraplegia [16,41], incomplete or permanent paraplegia [9,10,12,35,40,42,45,47,48], and even direct/indirect death [9,23,32,33,52]. Death is an uncommon consequence. It may result from spinal cord compression with neurological, respiratory, and/or cardiovascular impairment (direct) or from events such as pulmonary embolism that complicate a good neurological recovery after abscess/hematoma decompression (indirect) [52].

Most of the causes and consequences are associative, rather than causative. Neuraxial injuries are mostly linked to mechanical damage, drug-related neurotoxicity, or both. The response is usually inconsistent due to anatomo-physiological variations. In cases of tissue damage, the neurotoxicity increases due to the lack of protective connective tissue barriers. The use of vasoconstrictors can additionally complicate local anaesthetic clearance [6,19,59]. Accordingly, spinal cord ischaemia or vertebral canal haematoma have a markedly poor prognosis due to reduced blood flow, whereas meningitis and most of the nerve injuries and abscesses can fully recover [52,55]. However, if there is a delay in diagnosis, the prognosis is very poor [10,16]. The risk period for complications and related symptoms can be hours, days, or last for a week or more [36,39]. In adhesive arachnoiditis, the symptoms, including pain, are more nebulous and may take years to manifest [48]. Localized pain after epidural analgesia, usually at the waist, resulting from needling has also been reported, but it should be well distinguished from lower back pain [50].

The prevention of Anaes-SCI should heavily rely on personalization of the anaesthetic approach. A complete medical history followed by a careful patient examination is critical to adjusting the anaesthetic protocol and avoiding Anaes-SCI. Magnetic resonance imaging (MRI) is the preferred diagnostic modality to determine, for example, the pre-existence of spinal pathologies and the dimensions of the spinal canal and epidural space, the depth of which can be as small as 1.5 cm, even in adults. If MRI is not available, preoperative computerised tomography [52] can be considered, knowing the negative impact of radiation exposure, the poor accuracy of some diagnoses, and the lack of cost-effectiveness (only for specific patients). Nevertheless, it can provide the anaesthesiologist with invaluable data, mainly in high-risk patients or if the MRI is contra-indicated. The use of ultrasound, despite not being the first choice, may also help to determine the best approach when performing neuraxial blocks, particularly in patients with previous spine pathology and previous anaesthetic interventions [2,5,6,16,20,26,32,47].

During the perioperative period, it is crucial to verify possible signs of spinal cord trauma by searching for weakness or numbness and radicular back pain, as well as bowel and bladder dysfunction [46,47]. Blood pressure should also be evaluated. Unexplained hypotension can be a consequence of intrathecal injection of local anaesthetic during epidural analgesia associated with general anaesthesia, and the catheter position should be double-checked [8]. The level of puncture for the neuraxial technique varies depending on the specific area of the body that requires anaesthesia or analgesia for a particular patient. The epidural lumbar approach has fewer serious complications, particularly in higher-risk patients such as the elderly. The mid-thoracic spinal region is particularly susceptible to infarction due to its anatomically narrow canal, poor vascularization, and the presence of the Adamkiewicz spinal artery. Furthermore, the cervico-thoracic region has been reported as the more vulnerable and as with the highest risk of spontaneous epidural haematoma [34,35,36,37]. In fact, in the past, the thoracic approach was often avoided due to the fear of more serious complications from a haematoma or an abscess compared to the lumbar region. In what concerns subarachnoidal technique, despite the puncture location also being based on the desired area of the body that requires anaesthesia, the puncture is lumbar (always below L2 level). In this review, most of the complications were reported in the lumbar region. This fact is probably related to the highest frequency of neuroaxial techniques in this region.

In this context, to reduce the incidence of Anaes-SCI, continuous theoretical education and hands-on training to provide optimal technique with better pharmacological insight are of paramount importance [5,26,38,50,54,57]. It is important to respect the recommended patient position, to use a meticulous aseptic technique to remove the excess disinfectants that could irritate the spinal cord, to avoid large volumes of air in the resistance to air technique, and to proceed with needle withdrawal whenever pain is reported. To avoid anaesthetic neurotoxicity, the recommended concentrations and dosages must be carefully reviewed beforehand and respected. Additional care should be taken in the transforaminal and paramedian approaches to avoid vascular trauma [6,19,23,41,44,45,48,50]. Concerning epidural technique, if an accidental dural puncture happened, trying the epidural at the same spinal level should be avoided due to the risk of total spinal anaesthesia [60].

For post-technique monitoring, it is recommended to use postoperative surveillance to detect potentially treatable causes of neurological injury whenever necessary. It should be recalled that sometimes neurological signs are masked by a lack of patient consciousness, and an early diagnosis cannot be made [1,33,37]. It is important to exercise caution until the patient is discharged from the hospital. Although not as invasive as epidural catheter insertion, catheter removal can also lead to some of the complications already mentioned. Additionally, the timing of catheter removal in patients undergoing urgent antithrombotic therapy and cardiac complications should be postponed until heparin is discontinued and the platelet count and function are normal [34].

During the technique performance it is very important to be alert to any symptom occasionally encountered, such as pain or paraesthesia. Being alert does not mean to automatically give up but to re-evaluate and perhaps restart the procedure with another approach. However, during the continuous epidural analgesia, if sensory or motor loss is encountered, the pharmacologic administration must be immediately interrupted, and frequent evaluations should be conducted until signs of recovery are evident. Whenever signs of recovery failed in less than 1h, a multidisciplinary team that includes an anaesthesiologist, a neuroradiologist, a neurosurgeon and a neurologist should intervene, to prevent worsening of neurological symptoms. The first step is to do an emergent MRI to obtain the correct diagnosis. If a decompression is needed, it should be performed immediately due to tissue damage progression in time. Ideally, it should take place within 8 h post-technique/symptoms beginning, as recovery/outcome are a time-dependent. The pathophysiology and cellular changes occurring within the first 8 h after lesion are likely complex and appear to involve neurotoxic events due to the anaesthetic agent, along with damages derived from disruption of the blood-brain barrier. It should be noted, however, that pre-clinical studies have seldom focused on Anaes-SCI, compromising the full understanding of its pathophysiological mechanisms [5,6,21,37,61]. In this context, Acute Pain Units can individualise postoperative care, including post-operative surveillance, pain, diagnostic procedures, trauma, medical diseases, and complications related to the reported anaesthetic techniques [13].

The treatment is usually more related to the injuries and patient symptoms. For those with no evidence of neural deficit and mild symptoms or those whose punctures have been difficult, the follow-up includes assessment of vital signs, neurologic function and post-dural puncture headache. When symptoms linger, the neurologist/neurosurgeon should evaluate the patient, and neurophysiologic testing or imaging must be performed immediately. The follow-up of incomplete or unresolved lesions should be performed for up to 5 months [6,61,62]. In the reported cases included in this review, the most used drugs in Anaes-SCI treatment were antibiotics to treat abscess, corticoids to help cord decompression and analgesics for pain [8,10,33,35,40,41,43,44,45,48].

## 5. Conclusions

Since it is not feasible to perform randomized placebo-controlled clinical studies focusing on Anaes-SCI, the present systematic review evaluated only case reports, series of cases and epidemiological/clinical studies, which may be considered a limitation. We found that, despite the very low frequency, it is not possible to precisely determine the frequency and complications associated with Anaest-SCI. The reason for this resides in the very low number of reported cases, thus reducing the ability to fully understand and possibly correct underlying mechanisms and risk factors. In these conditions, it is difficult to propose firm recommendations [5,8]. Nevertheless, despite the very rare complications, neuraxial techniques are very important for proper pain prevention management, largely reducing patients’ morbidity, improving outcomes, reducing the length of hospital stay (enhanced recovery), and increasing pain chronification, with a consequent economic benefit. Another limitation of the present study is the exclusion of manuscripts written in languages other than English. This choice certainly reduced the misinterpretation of published material, and it did not impact the search and review of the studies included in this unique review.

In summary, neuraxial techniques are still one of the best options for opioid-sparing pain prevention and management. Even though Anest-SCI may occur, it is a very rare event that can be a consequence of many multifactorial components, including human knowledge and decisions, materials, equipment, drugs, and patient characteristics. Therefore, the present manuscript breaks ground, is, to our knowledge, one of the first systematic reviews in the field, and paves the way for more in-depth studies and the definition and/or improvement of anaesthetic protocols.

## Figures and Tables

**Figure 1 ijms-24-04665-f001:**
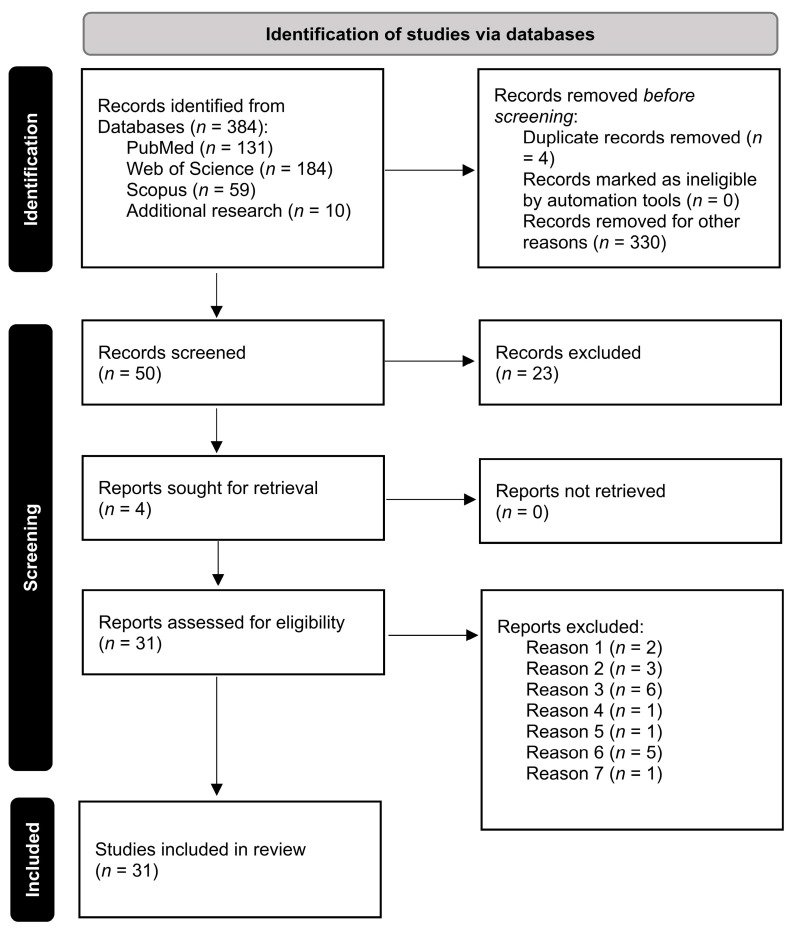
Selection process for studies included using PRISM diagram: http://www.prisma-statement.org (assessed on 12 October 2022).

**Table 1 ijms-24-04665-t001:** Summary of the main findings according to the cases reported in the selected manuscripts.

Ref	Patients/Age, RF	Type of Anaesthesia and Injury	Possible Reasons/Aetiology/ Consequences	Treatments/Conducts/ Recommendations
[3]	1 CR - Morquio Syndrome + stenosis	L2–3 epidural + GA	Spinal cord infarction leading to complete paraplegia	Epidural discontinued and catheter removal
[8]	4 paediatric CRs 23 months old 12 yo, BMI of 27 12 yo 11 yo	Epidural + GA: L3-4 - ischemia or venous hypertension 2 T12–L1 after 2 attempts - ischemia 3 L3-4 (blood returned on the first attempt) - inflamed nerve roots 4 T7-8 - vascular infarction	Flaccid paralysis, absent sensation MRI: ischemia or venous hypertension Paralysis, pain, absence of anal wink Urinary incontinence, numbness, and pelvic anaesthesia Pain, unresponsive and apnoeic, legs paralysis and paresis	Pharmacological and rehabilitation program with no further motor recovery Epidural catheter was removed, exploratory laminectomy Corticoids with full sensory and motor recovery Epidural discontinued, and catheter removed with permanent consequences
[9]	2 CRs 72 yo, hypertension 79 yo, no other RF found	L3-4 spinal - hematoma 2 L1-2 epidural - Hematoma	Complete paraplegia: one spinal cord compression (hematoma) and one subdural hematoma. A patient died in the decompression surgery	Drug discontinued and catheter removed + urgent decompression of the spinal canal
[10]	4 CRs 37 yo, alcohol abuse, trauma 28 yo, trauma 47 yo, trauma 49 yo, septicaemia	T8-9 epidural (10 days) - epidural abscess 2 L3-4 epidural (7 days) + T7-8 epidural removed due purulent secretion - epidural abscess 3 T7-8 epidural (17 days) - epidural abscess 4 T9-10 epidural (12 days) - epidural abscess	Severe pain, complete paraplegia after rehabilitation Incomplete paraplegia, bladder paresis, disturbed bowel function Incomplete paraplegia, paresis, numbness Incomplete paraplegia with impaired bladder and bowel control	Same treatment for all cases: epidural abscess evacuation + antibiotic treatment + rehabilitation
[11]	1 CR - 79 yo, diabetes	L2–L3 spinal - unilateral ischaemia	Severe subacute axonal sciatic damage and S1 root	Not reported
[12]	1 CR - 40 yo, BMI of 16.6	T12-L1 epidural - epidural bloody collection	Permanent paraplegia following percutaneous nephrolithotomy	Monitoring to allow early detection of mismanagement and prevention of further neurologic injury
[21]	1 CR - 28 yo, no RF found	L3-4 epidural - Spinal epidural hematoma with severe spinal cord stenosis	Pain, numbness, paraplegia, areflexia sensory loss and anal tone absent. Deep vein thrombosis	Surgical hematoma treatment and rehabilitation with functional recovery. Pharmacologic therapy to prevent further thrombosis
[23]	1 CR - 66 yo, 87 kg, thrombosis	Spinal anaesthesia with first attempt believed to be at the L3–4	Intense pain, paralysis, sensory deficit. Autopsy: extensive haematomyelia	Subarachnoid injection withdrawn and moved to GA
[34]	1 CR - 72 yo, systemic diseases (diabetes, hypertension…)	T11-12 epidural + GA - Spinal epidural hematoma	Fatigue in legs, loss of sensation, motor paralysis. CT + MRI showed a T9-11 spinal epidural hematoma	Emergency laminectomy and rehabilitation with symptoms slightly improved
[35]	1 CR - 81 yo, hypertension	T9-10 epidural + GA - Intracord catheterization + anaesthetic injection	Numbness, weakness, bowel, and bladder incontinence. Sensory loss below T11 and permanent paraplegia	Little improvement after corticoid and rehabilitation
[36]	1 CR - 34 yo, C3-4 trauma	C5-C6 epidural steroid block for pain control - posterolateral hematoma	Acute cervical myelopathy with pain, weakness	Hemilaminectomy with a near complete recovery
[37]	1 CR - 61 yo, no RF found	T10-11 epidural + GA - Infarction from conus to thoracic cord	Confusion, pyrexia and tachycardia. Systemic inflammatory response syndrome. L3 flaccid paralysis, areflexia, analgesia and impaired sensation	Epidural catheter removed and rehabilitation
[38]	1 CR - 69yo, recurrent pneumothoraces, angina	GA + several tentative of thoracic epidural - blood emerged from the needle	Spinal cord damage due to needle puncture and subsequent haematoma	Surgical dura repair with no improvement (paraplegic)
[39]	1 CR - 75yo, no other RF found	3 attempts of L2-L3 spinal anaesthesia - subdural hematoma	Mental confusion, fever, permanent paraplegia	Moved to GA.Antibiotic + antinflammatory + hematoma decompression
[40]	3 CRs 64 yo 68 yo 28 yo, drug addict	L3-4 epidural - hematoma 2 L3-4 epidural - epidural abscess 3. L1-2 epidural (pain) - Ischemia	Paraplegia due to epi-subdural hematoma Spastic paraplegia due epidural abscess Paracentral conus-epiconus ischemic lesion	Immediate laminectomy + rehabilitation Laminectomy + abscess removal Rehabilitation
[32]	1 CR - 83yo, heart disease	L1-2 epidural (paraesthesia) - Epidural haematoma	Limited sensory and motor function, bowel and bladder incontinent. 10 days later: gangrenous stump and septic shock	Urgent spinal cord decompression + rehabilitation
[41]	1 CR - 52 yo, 101 kg	L2-4 epidural (4 attempts) + GA - nerve root displacement due to extradural air	Prolonged paraesthesia and paresis	Corticoids. Patient with no pain or neurological symptoms
[42]	3 CRs 78 yo, no other RF found 30 yo, no RF found 29 yo, no RF found	L3-4 epidural - epidural haematoma 2 L3-4 epidural (pain and confusion) - presumptive anterior spinal artery syndrome 3 L3-4 epidural - anterior spinal artery syndrome	Motor and sensory loss Motor loss, loss of bladder and rectal sphincter function, numbness Pain and paralysis in the legs	Laminectomy with partial improvement Moved to GA + Neurologic consultations Neurologic consultations
[43]	1 CR - 7 yo, referred patient	T12–L1 epidural + propofol sedationunexpected needle puncture	Myodynamia improved, but progressive pain persisted that was absent after second treatment	Analgesics and corticoids, then neurotropin. Patient reported gradual pain decrease
[44]	2 CRs 40 yo, no RF found 41 yo, no RF found	Spinal (acute shock sensation) Spinal (electric shock sensation) - possible ischemia, trauma, neurotoxicity, and haemorrhage	Total loss of sensation. MRI revealed a T2 hyperintensity in right paramedian hemiconus Complete numbness and weakness below hip region in the left lower limb	Corticoids with no benefit. Neurorehabilitation with slow improvement Neurorehabilitation with partial improvement
[45]	1 CR - 21 yo, no RF found	L1-2 Spinal anaesthesia + T12–L1 interspace second attempt	Pain, persistent numbness, and weakness of her left lower limb with normal bladder and bowel sensations	Corticoids with gradual improvement
[46]	1 CR - 82 yo, ASA III, hypocoagulation	L4-5 epidural + GA + enoxaparin - epidural hematoma	2nd postoperative day reduced sensation of the right and motor weakness of the left leg	Laminectomy with no improvement in neurologic function
[47]	2 CRs 73 yo, 30.9 of BMI, ASA IV 39 yo, no RF found	T4-5 epidural + GA - epidural hematoma 2 T8-T9 epidural + GA - catheter malposition	MRI was performed only after paraplegia had developed the next day Paraesthesia, discomfort. Accidental re-start of epidural infusion led to coma, and respiratory arrest	Delayed hematoma evacuation with paraplegia Catheter position centrally in the spinal canal in CT
[33]	1 CR - 73 yo, hypertension	Attempted T11-12 epidural for pain management - spinal hematoma	Motor deficit on right lower limb. MRI showed a direct spinal cord injury	Pharmacological treatment and laminectomy with slow recovery
[32]	1 CR - 21 yo, no RF found	L3–4 spinal–epidural several attempts - Subdural hematoma	Left leg sensation and motor function completely recovered 3 h later	Hematoma absorption observation
[48]	1 CR - 27 yo, no RF found	L4-5 spinal - severe adhesive arachnoiditis	Pain, communicating hydrocephalus and syringomyelia. Rapid, progressive paraplegia and sphincter dysfunction	Unsuccessful laminectomy, external drainage of the syrinx and intravenous steroids
[49]	1 CR - 29 yo, no RF found	Combined spinal at L3-4 and epidural at L1-2 - adhesive arachnoiditis	Paraplegia, widespread syringomyelia, massive anterior arachnoid spinal cyst	Shunting of the cyst prevented symptoms progression. Numbness and motor weakness remained

Legend: Ref—reference, CR—case report, T—thoracic. L—lumbar, C—cervical, MRI—Magnetic Resonance Imaging, CT- Computed Tomography, GA—general anaesthesia, RF—risk factors, y—years old.

**Table 2 ijms-24-04665-t002:** Summary of the main findings according to the epidemiological clinical studies.

Ref	Study/Patients	Type of Anaesthesia	Anaes-SCI	Treatments/Conducts/ Recommendations
[13]	Retrospective Study: 10,838 referred to Acute Pain Unit	- neuraxial block - peripheral nerve block	10.1% with side effects/ complications: - 0.03% subcutaneous cell tissue hematoma - 0.03% epidural abscesses - 0.01% arachnoiditis - 0.08% peripheral nerve injury	The Acute Pain Units are fundamental in monitoring, following-up and guiding the treatment of patients with complications
[16]	Prospective study: 17,372 epidural catheters	- 67% epidural for perioperative pain relief - 22% for cancer pain - 11% for trauma-related pain	9 cases of epidural abscess: - 11% meningitis - 56% febrile - 67% local infection - 67% back pain - 78% neurologic disturbances	Main treatments: - intravenous antibiotic - neurosurgical decompression
[50]	Prospective study: 5083 surgical inpatients	- 80.5% lumbar - 19.5% thoracic	Major complications - 9.05% localized pain - 0.08% anaesthetic toxicity - 1.12% adverse neurological outcomes - 0.02% epidural hematoma	Anaesthesiologist’s skills could be improved to reduce the incidence of Anaes-SCI
[51]	Retrospective Study: 7958 non-obstetrical	- epidural anaesthesia	- 3.1% sanguineous puncture, - 1.6% accidental dural perforation - 0.94% unsuccessful catheter placement or insufficient analgesia	Increasing anaesthesiologists’ awareness of patients at higher risk for Anaes-SCI will enhance safety

Legend: Ref—reference.

## Data Availability

All data generated during this study is available upon reasonable request from the corresponding author.

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
