# Peer review of "Spinal Cord Injury and Complications Related to Neuraxial Anaesthesia Procedures: A Systematic Review"

_ijms, 2023, doi:10.3390/ijms24054665_

Round 1

Reviewer 1 Report

1.I would like to congratulate the authors for their effort to bring the available evidence in this important issue.  Very well organized review and very useful for anaesthetists, neurosurgeons and spinal surgeons. I would like to recommend the creation of a summarising table with all the possible reasons/ aetiology leading to spinal cord injury after neuroaxial anaesthesia procedures.

2. On the sentence (257): In  the mid-thoracic spinal region is particularly susceptible to infarction due to poor vascularization, I do think the authors should add the anatomical narrow canal in mid-thoracic spine and the presence of Adamkiewicz spinal artery.

3. The sentence : If a decompression is needed, it should take place within 8 hours 297 post-technique/symptoms beginning,..(297) I do think it will more prudent  to add the word immediately regrading the time of decompression and it will be more apprehensible to explain why the patient should be operated within 8 hours, explaining shortly the pathophysiology and the cellular changes which take place within 8 hours. 

Author Response

Dear Prof. Dr. Maurizio Battino

Editor-in-Chief of International Journal of Molecular Sciences,

Re: Ms. Ref. ijms-2218517

Thank you for the comments regarding our paper “Spinal Cord Injury and complications related to neuraxial an-aesthesia procedures: a systematic review” by Daniel H Pozza, Isaura Tavares, Célia Duarte Cruz and Sara Fonseca. We have reviewed the manuscript according to the critiques raised by the reviewers and thank them for their comments.

We hope you will consider the revised version of the manuscript suitable for publication in IJMS. All alterations in the manuscript are easily recognizable, since they are highlighted in yellow. A point-by-point answer to the issues raised by reviewers can be found below.

Reviewer 1

  1. I would like to congratulate the authors for their effort to bring the available evidence in this important issue.  Very well organized review and very useful for anaesthetists, neurosurgeons and spinal surgeons. I would like to recommend the creation of a summarising table with all the possible reasons/ aetiology leading to spinal cord injury after neuroaxial anaesthesia procedures.

Thank you very much for your time reviewing this manuscript, your kind comments, and your valuable suggestions. Related to your request, we clarified it by adding “possible reasons/aetiology” in the title of 3rd row (table 1) were you can find this information. Additionally, we also clarify it better in the text and it is summarized in lines 131-135 (page 4): “The possible reasons/aetiology of Anaes-SCI were: hematoma (14 cases), unspecified…”.

  1. On the sentence (257): “In the mid-thoracic spinal region is particularly susceptible to infarction due to poor vascularization,”I do think the authors should add the anatomical narrow canal in mid-thoracic spine and the presence of Adamkiewicz spinal artery.

We agree with you. This information has been added in page 10, line 259.

  1. The sentence : “If a decompression is needed, it should take place within 8 hours 297 post-technique/symptoms beginning,..(297)” I do think it will more prudent  to add the word immediately regarding the time of decompression and it will be more apprehensible to explain why the patient should be operated within 8 hours, explaining shortly the pathophysiology and the cellular changes which take place within 8 hours. 

We agree with you and we add the following information in page 11, lines 298-305:  “The pathophysiology and cellular changes occurring within the first 8 hours after lesion are likely complex and appear to involve neurotoxic events due to the anaesthetic agent, along with damages derived from disruption of the blood-brain barrier. It should be noted, however, that pre-clinical studies have seldom focused on Anaes-SCI, compromising the full understanding of its pathophysiological mechanicms”

Reviewer 2 Report

This is a very well conducted literature review of complications in neuraxial anesthesia procedures.  

In Figure one:  what are the "other reasons" that 330 papers were excluded?

In line 145.  You stated that corticoids were used to help decompression?  Is that true or is it for inflammation etc?

In line 305.  I think that neurosurgeons should likely be included in with neurologists about what to do if symptoms linger.

Author Response

Porto, 22nd February 2023

Dear Prof. Dr. Maurizio Battino

Editor-in-Chief of International Journal of Molecular Sciences,

Re: Ms. Ref. ijms-2218517

Thank you for the comments regarding our paper “Spinal Cord Injury and complications related to neuraxial an-aesthesia procedures: a systematic review” by Daniel H Pozza, Isaura Tavares, Célia Duarte Cruz and Sara Fonseca. We have reviewed the manuscript according to the critiques raised by the reviewers and thank them for their comments.

We hope you will consider the revised version of the manuscript suitable for publication in IJMS. All alterations in the manuscript are easily recognizable, since they are highlighted in yellow. A point-by-point answer to the issues raised by reviewers can be found below.

Reviewer 2

This is a very well conducted literature review of complications in neuraxial anesthesia procedures.  

Thank you very much for your time reviewing this manuscript, your kind comments, and your valuable suggestions.

In Figure one:  what are the "other reasons" that 330 papers were excluded?

The reasons were related to exclusion criteria, mainly those reported in lines 97-101: “Exclusion criteria: comments and editorials, only general anaesthesia, complications related to the spinal cord previous injury not related to the anaesthetic procedure, lesion by other needling causes such as acupuncture, treatment of chronic pain, metastasis in spinal cord due cancer discovered during anaesthetic techniques.” Other reason included the need to exclude duplicated studies.

In line 145.  You stated that corticoids were used to help decompression?  Is that true or is it for inflammation etc?

We agree with you and we modified the text accordingly.

In line 305.  I think that neurosurgeons should likely be included in with neurologists about what to do if symptoms linger.

We agree with you and we modified the text accordingly.